# Associations of Relative Humidity and Lifestyles with Metabolic Syndrome among the Ecuadorian Adult Population: Ecuador National Health and Nutrition Survey (ENSANUT-ECU) 2012

**DOI:** 10.3390/ijerph17239023

**Published:** 2020-12-03

**Authors:** Christian F. Juna, Yoon Hee Cho, Dongwoo Ham, Hyojee Joung

**Affiliations:** 1Department of Public Health, Graduate School of Public Health, Seoul National University, Seoul 08826, Korea; christian@snu.ac.kr; 2Facultad de Enfermería, Pontificia Universidad Católica del Ecuador, Quito 170525, Ecuador; 3Department of Biomedical and Pharmaceutical Sciences, The University of Montana, Missoula, MT 59812, USA; 4Institute of Health and Environment, Seoul National University, Seoul 08826, Korea; dwhampch@snu.ac.kr

**Keywords:** relative humidity, metabolic syndrome, physical activity, menopause, ENSANUT-ECU, 2012

## Abstract

The effects of the physical environment on metabolic syndrome (MetS) are still largely unexplained. This study aimed to analyze the associations of relative humidity of residence, lifestyles, and MetS among Ecuadorian adults. Data from 6024 people aged 20 to 60 years were obtained from an Ecuador national population-based health and nutrition survey (i.e., ENSANUT-ECU, 2012) and the mean annual relative humidity (%) from the Ecuador National Institute for Meteorology and Hydrology (2012). Odds ratio (OR) with 95% confidence intervals (CI) for MetS according to groups of relative humidity were calculated using multiple logistic regression. Living in high relative humidity (>80%) increased ORs of reduced high-density lipoprotein (HDL) cholesterol (1.25; 95 % CI, 1.06–1.56) and MetS (OR = 1.20; 95 % CI,1.01–1.42) in women. Furthermore, physically active men living in high relative humidity showed lower OR of elevated triglycerides (0.56; 95 % CI,0.37–0.85) while menopausal women living in high relative humidity showed increased ORs of MetS (5.42; 95 % CI, 1.92–15.27), elevated blood pressure (3.10; 95 % CI, 1.15–8.35), and increased waist circumference (OR = 1.34; 95 % CI, 1.09–1.63). Our results show that residence in high relative humidity and menopausal status increase ORs of MetS and its components in Ecuadorian women; however, physical activity significantly reduces the OR of elevated triglycerides in men. The obtained findings may help make public health policies regarding environmental humidity management, nutritional education, menopausal care, and physical activity promotion to prevent the onset of MetS among Ecuadorian adults.

## 1. Introduction

Metabolic syndrome (MetS) is comprised of elevated fasting glucose, hypertension, dyslipidemia, and abdominal obesity [1] and can lead to type 2 diabetes mellitus (DM2), heart disease, and cardiovascular disease (CVD) [2]. Over a billion people of the world population are estimated to have MetS [3]; in Latin American countries, a higher prevalence is observed, especially in women [4]. Human biology, lifestyle, and the environment have been found to promote the onset of MetS, though the specific causes are still unknown [5]. Like other Latin American countries, 31.2% of adults in Ecuador have MetS and approximately 85% of the total population have at least one of the MetS abnormalities [6]. Ischemic heart disease, DM2, and CVD were the top three causes of mortality in the Ecuadorian adult population in 2019 (3.1%, 6.5%, and 6.2% of the annual total mortality, respectively) [7]. Therefore, it is important to identify potential determinants of MetS development in Ecuador to set up strategies to minimize MetS onset as well as its subsequent diseases.

Biological factors (i.e., sex (female) and age (>40 years old)) and lifestyle patterns (i.e., smoking, alcohol consumption, unhealthy dietary habits, and physical inactivity) are recognized as risk factors for MetS development. In addition, physical environments, including elevation and climatic factors, have been reported to be associated with onset of metabolic diseases [8,9,10,11,12,13,14,15]. Among several physical environments, humidity is the most debated due to its inconsistent results in human health studies [16].

Humidity is linked to physiological responses through heat stress and hydration states [17]. Despite the importance of humidity on human body metabolism, it is rarely incorporated as an independent variable in many research studies. Rather, humidity is considered a confounding variable that may be related to the exposure and/or health outcomes [18]. Studies supporting the association of metabolic disorders and humidity are limited; however, a few studies showed that mortality in the USA tends to increase in hot and humid areas [19]. The less humid mountainous regions on the other hand showed the lowest prevalence of obesity (<30%) and MetS (30–35%) compared to the rest of the country [20]. Another study showed a positive association between diabetes mellitus, central obesity, higher systolic blood pressure, and lower physical activity in elder residents of the Mediterranean islands living in high relative humidity areas [21]. In addition, sociodemographic characteristics, epidemiological transition, globalization, and changes in lifestyle patterns (i.e., reduced physical activity and increased consumption of macronutrients and alcohol, and smoking) may modify these associations [6,22].

We hypothesized that relative humidity is associated with risk of MetS and that lifestyle patterns could modify the risk. To this end, we determined the associations of relative humidity, lifestyles, and dietary patterns with MetS among Ecuadorian adults based on data from the ENSANUT-ECU 2012 and the National Institute for Meteorology and Hydrology (INAMHI).

## 2. Methods

### 2.1. Study Design

A cross-sectional study was performed using ENSANUT-ECU, a nation-wide population-based health and nutrition survey conducted by the Ecuadorian Government and its Ministry of Public Health. Detailed explanations of the study design and data source profiles of ENSANUT-ECU are available elsewhere [23]. In brief, ENSANUT-ECU collected data that included sociodemographic characteristics of the population, housing, risk factors, food consumption, anthropometry, blood pressure, and nutritional biomarkers.

Written informed consent was obtained from all participants using consent forms, and a protocol of the study was approved by the Institutional Review Board of Seoul National University (code: SNU 19–04–003, date: 13 May 2019).

### 2.2. Subjects

A total of 11,044 participants over 20 years of age was initially included for ENSANUT-ECU with complete variables related to MetS. Among these individuals, we excluded 4166 rural residents who did not have physical activity data and other risk factors (*n* = 493) as well as participants who took medication for hypertension (*n* = 361). Thus, 6024 adult Ecuadorians (1964 men and 4060 women) were finally included in this study, as illustrated in Figure 1.

### 2.3. General Characteristics

The general information of participants, including age, sex, ethnicity, education level, and economic status of family, was collected from the housing questionnaire of ENSANUT-ECU 2012. Data regarding elevation of residence was also obtained. Ethnicity was classified as mestizo (Indian with European mix) and others (i.e., indigenous, montubio, afro-descendent, and white); education level was classified as primary, secondary, and college or higher; and economic status of family was classified as poor (Q1), middle (Q2 and Q3), and rich (Q4 and Q5). For menopause status, we subdivided women in two groups: menopausal stage “yes” for ≥50 years of age and “no” for <50 years old. This classification was based on the WHO parameters.

### 2.4. Lifestyles

Information regarding lifestyles was included as follows: current alcohol consumption and smoking (“yes” or “no”) were based on alcohol beverage intake and smoking in the past 30 days according to the United States National Survey on Drug Use and Health and WHO-Ecuador [24]. Physical activity was divided in two groups: “yes” for the performance of vigorous-intensity activity for at least 1 h 15 min, moderate-intensity activity for at least 2 h 30 min, or both for the past 7 days prior to the collection of data, and “no” for any activity that that took less than 1 h 15 min [25].

### 2.5. Anthropometric Measurements

Anthropometric measurements, including body mass index (BMI), height, weight, and waist circumference of participants, were performed at their residence by trained technicians using standardized procedures and portable equipment [26]. Blood pressure was measured twice using a sphygmomanometer according to standardized measurement techniques [26]; the mean of the readings was used in this study. Blood samples of total cholesterol, high-density lipoprotein (HDL) cholesterol, triglycerides, and glucose were collected from participants under an 8-h fasting period and measured using an enzymatic-colorimetric assay Modular Evo-800 (Roche Diagnostics, Mannheim, Germany). The Friedewald’s formula was used to calculate low-density lipoprotein (LDL) cholesterol [27]. Detailed explanations of the laboratory procedures are reported in the ENSANUT-ECU [23].

### 2.6. Dietary Intake

For dietary intakes, trained technicians collected data using the 1-day 24-h dietary recall method at the participants’ houses. The daily energy intake was calculated using the food composition table of the Food Dietary Guidelines for the Ecuadorian population (GABA) [28]. Analyses of calorie intake and estimated energy requirements (EER), according to age-, sex-, weight-, height- and physical activity level-specific equations were included in our study [29,30]. 

### 2.7. Relative Humidity

The mean annual relative humidity (%) of the urban area for 2012 was obtained from the National Institute for Meteorology and Hydrology INAMHI [31] and was used as a proxy for relative humidity at the participants’ cities of residence in order to assess the long-term association of ambient humidity with metabolic dysfunction. Areas above 80% were categorized as high relative humidity, and the rest were categorized as low-humidity areas. This categorization was based on the assumption that relative humidity above 80% is considered high and causes thermal discomfort and adverse health outcomes in hot-humid tropics [21,32,33], as in the case in Ecuador.

### 2.8. Metabolic Syndrome

The diagnosis of MetS was based on the harmonized guidelines of the National Cholesterol Education Program Adult Treatment Panel III and the Latin American Diabetes Association as the presence of three or more of the following components: waist circumference (men ≥ 94 cm and women ≥ 88), blood pressure (systolic ≥ 130 mmHg and/or diastolic ≥ 85 mmHg), HDL cholesterol (men < 40 mg/dL and women < 50 mg/dL), elevated triglycerides (≥150 mg/dL), or fasting glucose (≥100 mg/dL).

### 2.9. Statistical Analyses

All analyses were stratified by sex and relative humidity, with data presented as percentages or means with standard errors depending on the type of analysis. Differences in groups were compared using Chi-square and t-tests depending on the variable that was analyzed. Odds ratios (OR) and 95% confidence intervals (CIs) for MetS across the relative humidity groups were estimated using multiple logistic regression analysis. In the regression model, general and sociodemographic variables (age, ethnicity, economic status, and education level), anthropometric measurements (BMI, except for the model of waist circumference), lifestyles (physical activity, current alcohol consumption, current smoking, and total energy intake), elevation of residence, and menopausal status were considered. Statistical significance was defined as *p*-value < 0.05 in a two-tailed manner. Statistical analyses were performed using the PROC SURVEY procedures of SAS version 9.4 software (SAS Institute Inc., Cary, NC, USA) [34].

## 3. Results

The analyzed characteristics of the participants according to sex and relative humidity are shown in Table 1. The mean ages of participants were 34.6 ± 0.44 (SE) for men and 35.2 ± 0.35 years of age for women. A significant difference between groups was found in the following variables: ethnicity, economic status, education level, physical activity, elevation, and ambient temperature of residence (*p* < 0.05). However, age and smoking were significantly different in women only (*p* = 0.0332, *p* < 0.0001, respectively).

Table 2 shows that men had higher levels of blood pressure, total cholesterol, triglycerides, and energy intake, whereas women had higher BMI and EER (%). BMI was significantly different in women in the high relative humidity group only (*p* = 0.0461). Both sexes showed higher HDL cholesterol in high relative humidity (men *p* = 0.0278; women *p* = 0.0146). Additionally, in the high relative humidity sex groups, a significant increase of energy intake and EER (%) was observed (*p* < 0.05).

The prevalence and ORs for MetS according to relative humidity are presented in Table 3. Living in high relative humidity had effects on women only; they showed significant ORs of having reduced HDL cholesterol (1.25; 95% CI, 1.06–1.56) and MetS (OR = 1.20; 95% CI, 1.01–1.42) after adjusting for confounders.

Table 4 shows that current alcohol consumption and smoking enhanced increase of ORs of reduced HDL cholesterol and elevated triglycerides in women residing in high humidity (compared to Table 3). Interestingly, physically active men living in high relative humidity showed significantly lower ORs of elevated triglycerides (0.59; 95 % CI, 0.37–0.85).

Additionally, ORs for MetS in women according to their menopausal status and relative humidity of residence are presented in Table 5. Menopausal women living in high relative humidity showed increased ORs of MetS (5.42; 95 % CI, 1.92–15.27), elevated blood pressure (3.10; 95 % CI, 1.15–8.35), and increased waist circumference (OR = 1.34; 95 % CI, 1.09–1.63) compared to women without menopause.

## 4. Discussion

This study showed a positive association between relative humidity and MetS as well as reduced HDL cholesterol in women. Menopausal women living in high relative humidity showed higher ORs of MetS, elevated blood pressure, and increased waist circumference. Moreover, physically active men living in high relative humidity had lower OR of elevated triglycerides compared to physically inactive men living in the same humidity. To our knowledge, this is the first study to suggest that relative humidity is associated with MetS in an adult population using nationally representative data.

The prevalence of MetS in adult Ecuadorians shows similar trends with other Latin American countries and the USA [4,21]. However, higher OR of MetS in populations living in high relative humidity has never been reported before. An approximation to this finding could be observed in a study in the USA, where high relative humidity states presented a higher prevalence of MetS, obesity, and DM2 [21].

To further explore our findings, we performed additional analyses of MetS abnormalities, lifestyle patterns, and relative humidity. In women who reside in high relative humidity, who consume alcohol, and who smoke, higher ORs of reduced HDL cholesterol and physical inactivity increased the OR of MetS. These lifestyle factors could play moderating roles in MetS. Apart from the lack of previous studies on relative humidity and MetS, complementary research has investigated the effect of physical environment (i.e., humidity, temperature, elevation, radiation, etc.) on people’s health [19,35,36,37,38]. It was found that disparities in ambient temperature and relative humidity are associated with CVD [36], myocardial infarction morbidity and mortality [39,40], acute coronary syndrome [41,42], and DM2 [21].

Ecuador has four different geographical regions. The coast and the Galapagos Islands have the highest prevalence of MetS (35.0% and 41.9%, respectively), while the highlands and the Amazon have prevalence of 29.9% and 26.6%, respectively [6]. Each region has different environmental conditions and climatological patterns. According to Köppen’s climate classification, Ecuador has 11 different types of microclimates ranging from tropical to oceanic. Ecuador is situated on the equatorial line and thus produces little seasonality throughout the year: a warm rainy season lasts from January to April, with a cool and dry season from May through December [43]. The mean annual humidity in the coastal region is around 65%, while in the Amazon, it is around 85% [31]. The effects of high air humidity on human health are still controversial [44], and whether these climatological patterns may trigger MetS is difficult to answer here.

In order to identify a difference between sex, we analyzed sex and relative humidity individually. It was found that women living in high relative humidity had higher ORs of MetS and low HDL cholesterol. These pathophysiological differences may be explained by the effects of sexual hormones on the human body physiology [45,46]. The metabolism of lipids is modulated by endogenous sex hormones that might cause insulin resistance and abnormalities in the lipid profile [47]. Additionally, menopausal stage increased ORs of MetS, elevated blood pressure, and increased waist circumference in women living in high relative humidity compared to non-menopausal women living at the same humidity levels, indicating sex hormones may explain these differences. Several studies also revealed that menopause is associated with an increased risk of MetS and its components, which supports our outcomes [48,49,50]. 

Energy intake differed by sex and relative humidity (*p* < 0.05). This association suggested a regulatory role of hormones on appetite regulation and the onset of metabolic abnormalities. In contrast with our findings, other studies reported that exposure to high ambient temperatures and humidity under resting conditions caused poor appetite and reduced energy intake [51,52], which may be explained by some physiological factors such as reduced digestive enzyme activity and metabolic rate decrease [53].

Moreover, it has been proposed that an increase in relative humidity is associated with physical activity impairment [54,55]. Our findings indicate that physical activity is inversely associated with MetS and elevated triglycerides, similar to other studies that have shown a positive impact with exercise on cholesterol and triglyceride profile, abdominal obesity, and DM2 [56,57]. Several studies have described the benefits of physical activity on MetS and its components. However, the measurement of physical activity becomes difficult due to the use of different methods; thus, this should be considered when interpreting our results.

Alcohol consumption and current smoking were also associated with reduced HDL cholesterol in women living in high relative humidity areas. Some studies have shown that alcohol consumption has negative effects on the metabolism of glucose, cholesterol, and triglycerides [58,59]. In addition, smoking has been shown to alter the lipids and lipoprotein metabolism. Smokers have also shown lower adjusted levels of HDL cholesterol than nonsmokers [60,61].

This study must be interpreted considering the following limitations. First, among initial participants (*n* = 11,044), physical activity data was collected only for the urban population; thus, we excluded rural residents (*n* = 4166) and others (*n* = 854), resulting in a total of 6024 urban participants included for this study. Therefore, we cannot exclude the possibility of residual bias, and our results should be interpreted with caution. Second, casual inferences were excluded due to its cross-sectional design. Third, we could not infer the individual’s usual energy intake using a 1-day 24-h dietary recall. Forth, missing data on leptin and/or irisin concentration, which may help to better elucidate the association between humidity and MetS, is another limitation of the study. Therefore, further studies in the fields of human biology and environment are needed to determine the influence of relative humidity on MetS in the adult population. Despite these limitations, we utilized data from the ENSANUT-ECU 2012 which was conducted on a nation-wide scale across the entire country using standardized protocols and instruments and is the only survey the collected all the required biomarkers for diagnosis of MetS. Moreover, to our knowledge, this is the first study determining the association of relative humidity with MetS in a nationally representative population sample.

## 5. Conclusions

Our study suggests that living in high relative humidity (>80%) increases ORs of MetS and reduces HDL cholesterol in women. Furthermore, menopausal status enhances increase in OR of MetS and augments ORs of elevated blood pressure and increased waist circumference, while performing physical activity decreases the OR of elevated triglycerides in men. These findings come from nation-wide data and highlight the importance of management of relative humidity and lifestyles in Ecuadorians. Intersectoral programs aimed at controlling relative humidity in homes and work environments, providing nutritional education and menopause care, and increasing physical activity are needed to promote healthy life conditions and to prevent metabolic disorders.

## Figures and Tables

**Figure 1 ijerph-17-09023-f001:**
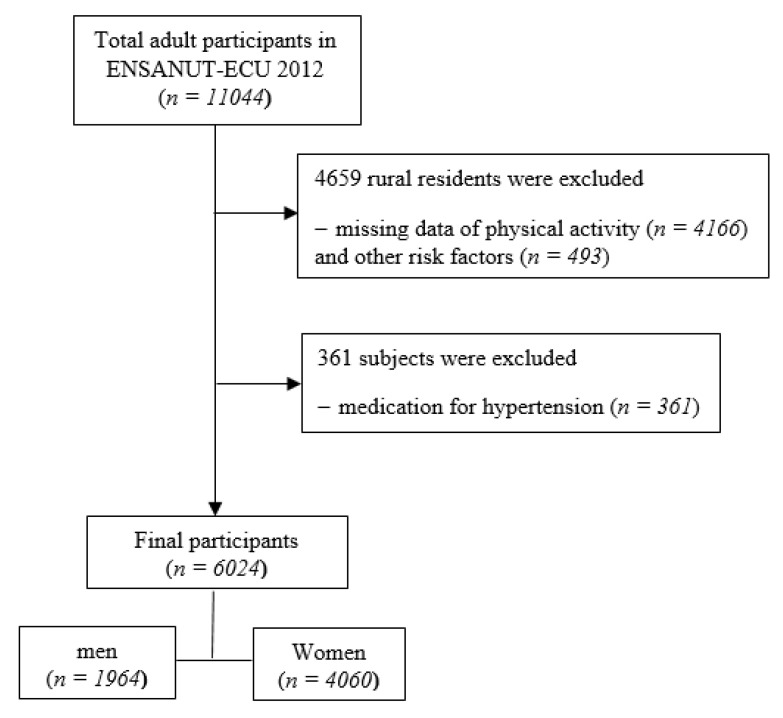
Flow diagram for the selection of study participants.

**Table 1 ijerph-17-09023-t001:** Descriptive characteristics of participants according to relative humidity of residence.

Variables	Men			Women		
	Low Relative Humidity	High Relative Humidity	*p*-Value	Low Relative Humidity	High Relative Humidity	*p*-Value
Number of people, N (%)	869 (44.2)	1095 (55.8)		1525 (37.6)	2535 (62.4)	
Age, years, N (%)			0.744			0.0332
20–29	305 (38.5)	400 (38.7)		498 (33.5)	923 (39.6)	
30–39	292 (29.6)	367 (28.7)		544 (30.3)	913 (30.0)	
40–49	198 (20.9)	249 (19.5)		397 (25.5)	595 (20.5)	
50–59	74 (11.0)	79 (13.1)		86 (10.7)	104 (9.9)	
Ethnicity, N (%)			<0.0001			0.0003
Mestizo	802 (88.2)	937 (73.4)		1397 (86.4)	2186 (79.0)	
Others	67 (11.8)	158 (26.6)		128 (13.6)	349 (21.0)	
Family economic status ^a^, N (%)			<0.0001			<0.0001
Low	191 (21.3)	342 (32.9)		400 (22.8)	835 (33.6)	
Middle	391 (44.4)	542 (49.6)		637 (40.6)	1245 (49.7)	
High	287 (34.3)	211 (17.5)		488 (36.6)	455 (16.7)	
Education level, N (%)			0.0004			<0.0001
Primary school	217 (22.9)	290 (28.9)		415 (24.3)	716 (32.6)	
Secondary school	395 (48.4)	569 (52.5)		694 (47.3)	1218 (45.4)	
College or higher	257 (28.7)	236 (18.6)		416 (28.4)	601 (22.0)	
Current alcohol consumption ^b^, N (%)			0.2684			0.9006
Yes	478 (56.6)	631 (59.9)		392 (25.7)	672 (25.4)	
No	391 (43.4)	464 (40.1)		1133 (74.3)	1863 (74.6)	
Current smoking ^c^, N (%)			0.1086			<0.0001
Yes	303 (30.9)	350 (26.5)		114 (8.1)	118 (3.5)	
No	566 (69.1)	745 (73.5)		1411 (91.9)	2417 (96.5)	
Physical activity ^d^, N (%)			0.0432			0.0007
Yes	394 (44.7)	456 (38.6)		276 (20.0)	400 (14.2)	
No	475 (55.3)	639 (61.4)		1249 (80.0)	2135 (85.8)	
Environmental conditions (mean ± SE)						
Elevation (masl)	1345.7 ± 58.6	703.1 ± 45.8	<0.0001	1456.8 ± 48.9	650.7 ± 28.7	<0.0001
Temperature (°C)	20.4 ± 0.2	22.6 ± 0.2	<0.0001	20 ± 0.2	22.7 ± 0.1	<0.0001

^a^ Low (Q1 and Q2), middle (Q3 and Q4), and high (Q5) according to the Wealth Index; ^b^ “yes” alcoholic beverage consumption in the past 30 days; ^c^ “yes” cigarette smoking in the past 30 days; ^d^ “yes” vigorous-intensity activities performance for at least 1 h 15 min or moderate-intensity activities for at least 2 h 30 min during the past seven days; masl, meters above sea level.

**Table 2 ijerph-17-09023-t002:** Anthropometry, biomarkers of metabolic syndrome (MetS), and energy intake by relative humidity of residence.

Variables	Low Relative Humidity	High Relative Humidity	*p*-Value	Low Relative Humidity	High Relative Humidity	*p*-Value
	(869)	(1095)		(1525)	(2535)	
Anthropometric and biochemical variable (mean ± SE)						
BMI (kg/m^2^)	26.5 ± 0.2	26.8 ± 0.2	0.4684	27.2 ± 0.2	27.6 ± 0.2	0.0461
Waist circumference (cm)	94.7 ± 2.2	95.6 ± 2.4	0.7722	98.3 ± 3.7	91.5 ± 1.3	0.0796
SBP (mmHg)	125.1 ± 2.1	123.9 ± 1.7	0.6468	116.3 ± 1.3	115.6 ± 0.9	0.6341
DBP (mmHg)	79.9 ± 2.2	79.0 ± 1.7	0.7634	73.7 ± 1.3	72.3 ± 0.8	0.3956
Fasting glucose (mg/dL)	93.8 ± 1.2	95.2 ± 1.9	0.5249	92.7 ± 0.8	93.0 ± 1.1	0.8312
Total cholesterol (mg/dL)	186.7 ± 1.8	184.8 ± 1.8	0.4416	181.4 ± 1.3	178.3 ± 1.3	0.0899
HDL cholesterol (mg/dL)	40.6 ± 0.5	42.2 ± 0.6	0.0278	45.5 ± 0.5	47.0 ± 0.4	0.0146
LDL cholesterol (mg/dL)	112.2 ± 1.5	110.4 ± 1.5	0.3928	108.9 ± 1.1	108.7 ± 1.0	0.8622
Triglyceride (mg/dL)	179.2 ± 5.5	169.4 ± 6.7	0.2589	129.7 ± 3.2	122.2 ± 2.3	0.0541
Macronutrient intake (mean ± SE)						
Energy (kcal)	2123.8 ± 20.2	2210.5 ± 20.8	0.0028	1826.9 ± 15.1	1868.5 ± 14.3	0.0463
EER (%)	81.6 ± 1.0	86.1 ± 1.0	0.0033	97.1 ± 0.9	99.8 ± 0.9	0.0364

BMI, body mass index; DBP, diastolic blood pressure; HDL, high-density lipoprotein; LDL, low-density lipoprotein; SBP, systolic blood pressure; EER, estimated energy requirement.

**Table 3 ijerph-17-09023-t003:** Prevalence and adjusted odds ratio (95% confidence intervals) for MetS by relative humidity of residence.

		Men		Women
Components of MetS	Low Relative Humidity	High Relative Humidity	Low Relative Humidity	High Relative Humidity
	(*n* = 869)	(*n* = 1095)	(*n* = 1525)	(*n* = 2535)
Increased waist circumference				
Prevalence (%)	34.00	19.17	41.14	25.62
OR (95 % CI)	1.00 (ref)	1.13 (0.85–1.48)	1.00 (ref)	1.21 (0.95–1.53)
Elevated blood pressure				
Prevalence (%)	18.50	9.99	8.85	5.35
OR (95 % CI)	1.00 (ref)	0.96(0.71–1.28)	1.00 (ref)	1.02 (0.75–1.38)
Reduced HDL cholesterol				
Prevalence (%)	33.34	16.33	25.55	38.69
OR (95 % CI)	1.00 (ref)	0.87 (0.66–1.13)	1.00 (ref)	1.25 (1.06–1.56)
Elevated triglycerides				
Prevalence (%)	30.98	14.22	16.96	9.30
OR (95 % CI)	1.00 (ref)	0.80 (0.61–1.06)	1.00 (ref)	0.87 (0.68–1.11)
Elevated fasting glucose				
Prevalence (%)	9.24	7.13	9.31	5.73
OR (95 % CI)	1.00 (ref)	1.20 (084–1.70)	1.00 (ref)	0.83 (0.61–1.12)
Metabolic syndrome				
Prevalence (%)	24.13	12.41	11.53	17.73
OR (95 % CI)	1.00 (ref)	0.84 (0.61–1.14)	1.00 (ref)	1.20 (1.01–1.42)

All values accounted for the complex sampling design effect using the PROC SURVEY procedure. OR, odd ratio; CI, confidence interval; HDL, high-density lipoprotein. The multiple logistic regression analysis was adjusted for age, ethnicity, family economic status, education level, BMI (except for the model of waist circumference), physical activity, alcohol consumption, smoking, energy intake, and resident elevation.

**Table 4 ijerph-17-09023-t004:** Odds ratios for MetS among residents living in high relative humidity according to lifestyle factors.

Components of MetS	Men				Women			
	Yes (95 % CI)	*p*-Value	No (95 % CI)	*p*-Value	Yes (95 % CI)	*p*-Value	No (95 % CI)	*p*-Value
Current alcohol consumption								
Increased waist circumference	1.18 (0.81 ± 1.73)	0.3817	0.99 (0.65 ± 1.51)	0.9757	1.58 (0.99 ± 2.51)	0.0552	0.93 (0.70 ± 1.25)	0.6386
Elevated blood pressure	1.09 (0.78 ± 1.53)	0.6478	1.03 (0.69 ± 1.54)	0.9025	0.83 (0.50 ± 1.37)	0.1115	1.02 (0.78 ± 1.34)	0.8161
Reduced HDL cholesterol	0.86 (0.58 ± 1.27)	0.4400	0.77 (0.48 ± 1.23)	0.2400	1.31 (1.01 ± 1.69)	0.0350	0.99 (0.65 ± 1.53)	0.9746
Elevated triglycerides	0.76 (0.52 ± 1.10)	0.1464	0.78 (051 ± 1.19)	0.2487	0.88 (0.49 ± 1.56)	0.4711	1.03 (0.80 ± 1.34)	0.8119
Elevated fasting glucose	1.09 (0.70 ± 1.73)	0.6922	1.60 (0.94 ± 2.70)	0.0785	0.54 (0.27 ± 1.06)	0.0735	0.95 (0.68 ± 1.33)	0.7663
Metabolic syndrome	0.87 (0.60 ± 1.32)	0.4988	0.88 (0.54 ± 1.41)	0.7616	0.79 (0.47 ± 1.33)	0.3849	1.16 (0.87 ± 1.54)	0.3078
Current smoking								
Increased waist circumference	1.24 (0.78 ± 1.96)	0.3684	1.19 (0.86 ± 1.64)	0.3030	1.43 (0.90 ± 2.28)	0.1290	0.56 (0.21 ± 1.50)	0.2464
Elevated blood pressure	0.56 (0.36 ± 0.87)	0.0101	0.87 (0.61 ± 1.25)	0.4605	0.89 (0.27 ± 2.95)	0.8537	0.91 (0.69 ± 1.19)	0.4910
Reduced HDL cholesterol	0.89 (0.57 ± 1.38)	0.5916	0.99 (0.73 ± 1.36)	0.9612	1.95 (1.30 ± 2.93)	0.0013	1.08 (0.83 ± 1.39)	0.5484
Elevated triglycerides	1.00 (0.63 ± 1.59)	0.9877	0.78 (0.56 ± 1.16)	0.1177	0.41 (0.15 ± 1.07)	0.0681	1.07 (0.86 ± 1.33)	0.5359
Elevated fasting glucose	1.21 (0.62 ± 2.36)	0.5720	0.88 (0.60 ± 1.29)	0.5017	0.66 (0.12 ± 3.60)	0.6278	0.81 (0.61 ± 1.08)	0.1480
Metabolic syndrome	1.03 (0.64 ± 1.65)	0.9118	1.05 (0.67 ± 1.66)	0.7658	0.35 (0.11 ± 1.05)	0.0623	1.09 (0.82 ± 1.47)	0.4500
Physical activity								
Increased waist circumference	0.98 (0.60 ± 1.61)	0.9300	1.37 (0.85 ± 2.20)	0.1937	0.83 (0.44 ± 1.58)	0.5655	1.15 (0.88 ± 1.47)	0.2747
Elevated blood pressure	1.04 (0.70 ± 1.55)	0.7428	1.13 (0.80 ± 1.60)	0.4095	0.69 (1.05 ± 1.13)	0.3457	1.06 (0.78 ± 1.43)	0.7264
Reduced HDL cholesterol	0.94 (0.64 ± 1.38)	0.7484	0.86 (0.58 ± 1.26)	0.4387	1.28 (0.77 ± 2.12)	0.3349	1.20 (0.97 ± 1.48)	0.0975
Elevated triglycerides	0.59 (0.37 ± 0.85)	0.0067	0.94 (0.64 ± 1.38)	0.7556	0.88 (0.49 ± 1.59)	0.6650	1.01 (0.80 ± 1.28)	0.9145
Elevated fasting glucose	1.16 (0.65 ± 2.08)	0.6210	0.84 (0.55 ± 1.27)	0.4022	0.96 (0.42 ± 2.24)	0.9325	0.79 (0.59 ± 1.06)	0.1125
Metabolic syndrome	0.87 (0.54 ± 1.45)	0.5708	0.79 (0.52 ± 1.22)	0.2860	1.03 (0.55 ± 1.91)	0.9368	1.23 (1.02 ± 1.48)	0.0301
	**High ^a^** **(95 % CI)**	***p*-Value**	**Low** **(95 % CI)**	***p*-Value**	**High ^a^** **(95 % CI)**	***p*-Value**	**Low** **(95 % CI)**	***p*-Value**
% EER								
Increased waist circumference	0.91 (0.45 ± 1.87)	0.8163	1.24 (0.85 ± 1.80)	0.2672	1.35 (0.98 ± 1.86)	0.0645	0.85 (0.60 ± 1.18)	0.3307
Elevated blood pressure	1.06 (0.52 ± 2.15)	0.6800	1.11 (0.81 ± 1.52)	0.5308	1.18 (0.78 ± 1.78)	0.5875	0.96 (0.71 ± 1.32)	0.3117
Reduced HDL cholesterol	0.96 (0.46 ± 1.99)	0.4993	0.77 (0.53 ± 1.13)	0.2813	1.06 (0.77 ± 1.63)	0.6920	1.32 (0.99 ± 1.79)	0.0578
Elevated triglycerides	0.86 (0.39 ± 1.87)	0.8164	0.82 (0.50 ± 1.06)	0.1985	0.88 (0.66 ± 1.64)	0.4841	1.05 (0.80 ± 1.41)	0.7065
Elevated fasting glucose	0.88 (0.43 ± 1.80)	0.7201	1.01 (0.68 ± 1.56)	0.9564	0.85 (0.55 ± 1.30)	0.4597	0.77 (0.53 ± 1.11)	0.1605
Metabolic syndrome	0.81 (0.33 ± 1.92)	0.6444	0.91 (0.65 ± 1.28)	0.5948	0.91 (0.64 ± 1.29)	0.6005	1.22 (0.91 ± 1.65)	0.1808

ORs were calculated based on low relative humidity residents (reference); ^a^ high (>100%) or low (≤100%) for EER; CI, confidence interval; HDL high-density lipoprotein; EER, estimated energy requirement.

**Table 5 ijerph-17-09023-t005:** Odds ratios for MetS among women living in high relative humidity according to menopausal stage.

Components of MetS	Women (*n* = 4060)			
	Yes (95 % CI)	*p*-Value	No (95 % CI)	*p*-Value
Menopausal stage				
Increased waist circumference	1.34 (1.09 ± 1.63)	0.0045	0.22 (0.02 ± 3.04)	0.2550
Elevated blood pressure	3.10 (1.15 ± 8.35)	0.0253	0.80 (0.60 ± 1.07)	0.1382
Reduced HDL cholesterol	1.48 (0.59 ± 3.74)	0.4064	1.13 (0.93 ± 1.37)	0.2243
Elevated triglycerides	2.37 (0.96 ± 5.90)	0.0627	0.87 (0.70 ± 1.08)	0.2130
Elevated fasting glucose	0.90 (0.31 ± 2.64)	0.8510	0.81 (0.61 ± 1.08)	0.1442
Metabolic syndrome	5.42 (1.92 ± 15.27)	0.0015	0.89 (0.71 ± 1.12)	0.3177

ORs were calculated based on low relative humidity residents (reference). CI, confidence interval; HDL high-density lipoprotein.

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
