# Peer review of "Associations of Relative Humidity and Lifestyles with Metabolic Syndrome among the Ecuadorian Adult Population: Ecuador National Health and Nutrition Survey (ENSANUT-ECU) 2012"

_ijerph, 2020, doi:10.3390/ijerph17239023_

Round 1

Reviewer 1 Report

Dear authors,           

Please explain the novelty of the results. The data calculated were obtained from the ENSANUT-ECU  (2010-2012). Over the past 10 years, the world has changed significantly (Ecuador’s GDP rose from 69.56 in 2010 to 107 in 2019-2020, the political changes as well as cultural were also present). Based on this, it is necessary to explain why the presented data are still relevant.

Reviewer 2 Report

REVIEW - "Associations of Relative Humidity and Lifestyles with Metabolic Syndrome Among the Ecuadorian Adult Population: Ecuador National Health and Nutrition Survey (ENSANUT-ECU) 2012"

That is a very intersting and robust research about relative humidity levels factor and MetS. This work certainly would be very useful for the development and reflection of health strategical issues (it has very interesting data into a still "nebulous", doubtful theme).

Lines 78-79: "The Institutional Review Board of Seoul National University approved the present study (code: 78 SNU 19-04-003). All participants signed an informed form". - I think that there is a mistake here. Please check.

Lines 253-254: "Our study showed that living in high relative humidity (> 80%) increased OR of reduced HLD cholesterol in women ..." - change HLD to HDL.

CONCLUSIONS: This sector certainly could be improved. Let's develop this conclusion sector.

Very good work at all.

Reviewer 3 Report

This interesting study has examined the associations of relative humidity and lifestyle factors with metabolic syndrome in Ecuador. The findings are intriguing. Several concern were raised when I read the manuscript.

  1. Lines 81-83: As the original sample size of this nationally representative survey was 11,044, and 5,020 residents were excluded in the present study. Would the authors please add a table presenting the possible differences in the characteristics of the included and excluded individuals?
  2. It would be nice if the authors could clarify why they used 80% as the cut-off point to categorize relative humidity; also is it worthwhile to consider relative humidity as a continuous variable in the modelling?
  3. As the authors have mentioned, a bunch of environmental factors are associated with metabolic syndrome (e.g., cold temperature). The authors may have thought about it, but is it worth it to consider controlling for ambient temperature in the model?
  4. Line 131: I assume ‘temperatures’ here was supposed to be ‘relative humidity’.

Reviewer 4 Report

The authors focused their attention on an interesting topic: the possible correlation between the risk of metabolic syndrome and humidity. The possibility of exploiting Ecuador's environmental heterogeneity has helped and enriched their research. But in any case, they speculate that humidity is associated with a different hunger / satiety control.
The authors could enrich their work by measuring at least leptin concentration in order to bolster their hypothesis. Furthermore, given that the role played by exercise has also been analyzed, it could be useful to dose irisin, which as known is not only secreted following physical exercise but promotes thermogenesis and positively influences insulin resistance. They might have seen large numbers even doing these analyzes on a smaller sample.
Furthermore, the same authors discuss the hormonal influence. Since the analyzed cohort is composed for the most part by women, can the authors subdivide the group of women not only by age group but by fertile age and menopause?

Round 2

Reviewer 4 Report

I accept in this form